

# A low activity ion source for measurement of atmospheric gases by CIMS

Young Ro Lee[1], Yi Ji[1], David J. Tanner[1], L. Gregory Huey[1]

[1]School of Earth and Atmospheric Sciences, Georgia Institute of Technology, Atlanta, 30332, USA

*Correspondence to*: L. Gregory Huey (greg.huey@eas.gatech.edu)

**Abstract.** Most I⁻-CIMS (iodide-chemical ionization mass spectrometers) for measurement of atmospheric trace gases utilize a radioactive ion source with an initial activity of 10 or 20 mCi of $^{210}$Po. In this work, we characterize a $^{210}$Po ion source with an initial activity of 1.5 mCi that can be easily constructed from commercially available components. The low level of radioactive activity of this source significantly reduces complications with storage and shipping relative to higher activity

sources. We compare the sensitivity of the low activity source (LAS) to a standard 20 mCi source, as a function of carrier gas flow and flow tube pressure, for peroxyacetyl nitrate (PAN), formic acid ($HCO_2H$) molecular chlorine ($Cl_2$), and nitryl chloride ($ClNO_2$) using an I⁻-CIMS. The LAS provides 2 to 5 times less sensitivity than that of the standard source even though the ratio of activity is approximately 13. However, detection limits of less than 2 pptv for the tested compounds are achieved for integration times of the order of a minute. The sensitivity of the LAS is less dependent on the magnitude of the carrier gas than

a standard source. In addition, the LAS provides maximum sensitivity at relatively low carrier gas flows. Finally, we demonstrate that the LAS can be used to measure PAN in the remote atmosphere from an aircraft by showing data obtained on the NASA DC-8 during the Atmospheric Tomography (ATom) mission. In summary, the LAS may be an excellent substitute for a standard ion source in some field applications.

## 1 Introduction

20       Chemical ionization mass spectrometry (CIMS) is a versatile method for the measurement of atmospheric trace gases (Huey et al., 2007). CIMS has excellent sensitivity and rapid time response (t<1 s) which allows the observation of atmosphere compounds with high temporal resolution which is particularly important for airborne measurements. There are a variety of different CIMS methods used to measure atmospheric species such as: PTR-MS (Proton-Transfer Mass Spectrometry), atmospheric pressure CIMS commonly utilizing $NO_3^-$ as a reagent ion, reduced pressure CIMS utilizing I⁻ as well as other

reagent ions such as $CF_3O^-$, acetate, and $SF_6^-$ (Lindinger et al., 1998; de Gouw and Warneke, 2006; Eisele and Tanner, 1993; Huey et al., 1995, 1996; Crounse et al., 2006; Veres et al., 2008; Bertram et al., 2011; Slusher et al., 2001).

      The defining characteristic of CIMS is ionization of a compound by a chemical reaction with a reagent ion. Reagent ions are produced by flowing precursor gases through an ion source. Typically, most I⁻-CIMS flow dilute methyl iodide in



nitrogen through a tube containing a $^{210}$Po metal foil which emits alpha particles. The alpha particles ionize the nitrogen

producing secondary electrons that can be thermalized and dissociatively attached by methyl iodide to form I$^-$. PTR-MS instruments typically use a hollow cathode electrical discharge to produce $H_3O^+$ ions that can be introduced into a drift tube reactor (de Gouw et al., 2003b; Inomata et al., 2006). Other common ion sources include X-ray, Americium (alpha emitter), and corona discharges (Jost et al., 2003; Kürten et al., 2011; Zheng et al., 2015). The choice of an ion source for a specific application is influenced by many factors that include: operating pressure, ion current, background contamination, and

regulatory environment. For example, radioactive ion sources are generally not used at operating pressures below 10 torr as the path length of the emitted alpha particles is too large to generate sufficient ion current. In contrast, PTR-MS typically operate at ionization pressures below 1 torr which is more compatible with a discharge source (Yuan et al., 2017).

The use of radioactive ion sources for CIMS has several significant advantages. These include stability, ease of use, and reliability. Perhaps the most important feature is the efficiency of ion generation. A 20 mCi $^{210}$Po ion source produces 7.4

x 10$^8$ disintegrations per second. This dissipates less than 1 mW of power, and can produce more than 10$^7$ reagent ions per second impacting an ion detector (Nah et al., 2018). Thus, high ion count rates are achieved while limiting the production of radicals that can react and produce molecules that interfere with measurement of atmospheric species.

There are significant disadvantages to radioactive ion sources as well. Polonium sources need to be replaced regularly, due to a half-life of 138 days, at some expense. More importantly polonium is highly toxic and most $^{210}$Po sources require

HAZMAT shipping and a significant degree of institutional oversight. For example, standard radioactive sources used in CIMS are commercial anti-static devices, P-2021 (10 mCi) and P-2031 (20 mCi), manufactured by NRD. Both of these sources require HAZMAT shipping if returned to the manufacturer on the time scale of months. The activity level of these sources may constrain their use in field locations for which permissions to transport or use radioactive source may not be feasible (49CRF 173.425, Table 4). In addition, while these commercially available devices are designed to be used in evacuated

systems, they may require additional preparation to ensure a good vacuum seal. For example, we have found that a significant fraction of the 20 mCi ionizers have leaks on the body of the device.

One problem that we have repeatedly encountered at remote field sites or at aircraft facilities is a lack of ability to store the radioactive sources which necessitates shipping the source back to the manufacturer immediately after the field observations are completed. However, most field campaigns are of the order of a month or two in duration which necessitates

HAZMAT shipping of the standard sources. The goal of this work is to develop a source with lower activity that we can easily ship back to the manufacturer after field use. In this manuscript, we describe a $^{210}$Po ion source with an initial activity of 1.5 mCi using standard anti-static products, commonly employed in mass balances, mounted in commercially available vacuum fittings, which can be easily interchanged with traditional radioactive ion sources. This initial level of activity for the LAS does not require specific packaging, labelling, and documentations except for the UN identification number (i.e., UN2911)

marked on the outside of the package and included in the packing list (49CFR 173.410; 49CRF 173.425, Table 4). We compare



the sensitivity of this source to a standard 20 mCi source for CIMS measurements of peroxyacetyl nitrate (PAN), molecular chloride ($Cl_2$), nitryl chloride ($ClNO_2$), and formic acid ($HCO_2H$). We also show the performance of the LAS for airborne measurement of PAN from the NASA DC-8 research aircraft during the Atmospheric Tomography (ATom) mission.

## 2 Method

### 2.1 Low activity ion source


A drawing and photograph of the LAS are shown in Figure 1. The LAS contains three commercial anti-static ionizing cartridges (2U500, NRD), housed in a standard QF-40 nipple. Each cartridge has an initial activity of 500 $\mu Ci$ equivalent to $1.85 \times 10^7$ disintegration per second and is equipped with a protective grid to prevent physically touching of the polonium foil (Figure 1b). The cartridges, with a total activity of 1.5 mCi (55.5 MBq), were mounted on a thin (~0.1 mm) stainless-steel foil that is

6 cm wide and 10 cm height as shown in the left of Figure 1. The foil with the cartridges were folded to fit in a QF-40 nipple. The QF-40 stainless steel nipple containing ionizing cartridges was connected to a flow tube reactor (QF-40 stainless steel tee). The opposite opening of the nipple was connected to a manifold that enables delivery of a flow of dilute methyl iodide ($CH_3I$) in nitrogen.

### 2.2 Laboratory configuration for TD-CIMS PAN measurement technique

Characterization of the LAS is performed using the TD-CIMS technique for PAN measurement (Slusher et al., 2004; Zheng et al., 2011). The CIMS instrument used herein is very similar to that described by Kim et al., (2007) and Chen et al., (2016). As shown in Figure 2, the CIMS consists of the following differently pumped regions: a flow tube, a collisional dissociation chamber (CDC), an octopole ion guide and a quadrupole mass filter with an ion detector. Ambient air is sampled through an inlet constructed from 1.27 cm outer diameter and 0.95 cm inner diameter FEP Teflon tubing at a constant flow rate of

approximately 2.7 standard liters per minute (slpm). Background measurements were performed regularly by drawing ambient air through a QF 40 nipple filled with stainless steel wool heated to 150 C, which effectively destroys the thermally generated radicals (Flocke et al., 2005; Zheng et al., 2011). The PAN calibration standard is generated from a photolytic source similar to that described by Warneck and Zerbarch (1992). The output of the photolytic source is periodically added to PAN free air, downstream of the heated metal tubing. A 29.6 cm section of the Teflon tubing in front of the flow tube orifice is heated to an

external temperature of approximately 180℃ to thermally dissociate PAN (R1). It should be noted that longer length and higher temperature than typical operation conditions for PAN measurements are used to achieve full thermal dissociation of PAN within the residence time in the heated inlet. Additional details are provided in Figure S1.

$$RC(O)O_2NO_2 \xrightarrow{\Delta} RC(O)O_2 + NO_2, \quad \text{(R1)}$$

$$RC(O)O_2 + I^- \cdot (H_2O)_n \rightarrow RC(O)O^- \cdot (H_2O)_n + IO, \quad \text{(R2)}$$


$RC(O)O^- \cdot (H_2O)_n + M \rightarrow RC(O)O^- + H_2O,$ (R3)

The thermal dissociation (TD) region is evacuated into the CIMS and a small scroll pump (Varian IDP3; 60 L min$^{-1}$). The TD region was maintained at a constant pressure of 100 torr by the combination of an upstream metering valve and a commercial pressure controller (MKS 640) mounted downstream of the TD before the small scroll pump. Reduced pressure in the thermal dissociation region minimizes negative interferences (e.g. loss of sensitivity) due to NO, NO$_2$ and radical-radical reactions.

The impact of ambient NO and NO$_2$ on sensitivity was minimal in these experiments as shown in SI Figure S1b. The flow rate before the pressure controller was held constant at 2.7 slpm so a constant dilution of the calibration standard and thermal dissociation conditions were maintained. The sample mass flow into the low-pressure flow tube is controlled by an automatic variable orifice (AVO). The area of a triangular opening in the AVO varies by a microprocessor controlled stepping motor. The AVO is used in these experiments to maintain a constant total mass flow into the flow tube as the ion source flow is varied.

This feature has been utilized in Chen et al., (2016) to maintain a constant flow tube pressure over the vertical range of aircraft sampling. An example of the gas flows at a flow tube pressure of 40 torr is shown in Figure S1c.

The experiments were carried out using two different scroll pumps (Edwards nxDS 6i or Varian Triscroll$^{TM}$ 300) with corresponding pumping speeds of 105 L min$^{-1}$ and 240 L min$^{-1}$. This allowed us to evaluate the use of the LAS under conditions typical of both airborne and ground based operation. We commonly use a large scroll pump for ground-based measurements

but are often required to use a smaller scroll pump due to power and weight limitations encountered on aircraft. Flow tube pressures higher than 20 torr were achieved by placing a restrictive orifice between the flow tube and the scroll pump. The total flows sampled into the scroll pumps were 2.7 and 6.5 slpm, respectively. Iodide reagent ions (I$^-$) are produced by passing dilute methyl iodide in nitrogen through a tube containing $^{210}$Po (1.5 mCi or 20 mCi). Ion-molecule reactions occur over the length of the flow tube (R2-R3), and the resulting product ions are transmitted into a collisional dissociation chamber (CDC)

evacuated by a molecular drag pump (Alcatel MDP 3011). Weakly bound cluster ions are dissociated into core ions in the CDC (e.g., $RC(O)O^- \cdot (H_2O)_n$ dissociates into $RC(O)O^-$). The ions are guided by octopole to the quadrupole filter for mass selection and then detected by an electron multiplier. PAN is detected as acetate anion (59 m/z $^{12}CH_3^{12}C(O)O^-$ or 61 m/z $^{13}CH_3^{13}C(O)O^-$) in the TD-CIMS.

## 2.3 Laboratory configuration for Iodide-Adduct CIMS

In addition to the characterization of the LAS using the TD-CIMS technique, similar experiments are performed to demonstrate the capabilities of the LAS using iodide-adduct formation for the detection of formic acid, Cl$_2$ and ClNO$_2$. This tests the LAS for a more generally used chemical ionization method (e.g. Le Breton et al., 2012; Lee et al., 2014; Osthoff et al., 2008; Thornton et al., 2010; Mielke et al., 2011). The configuration of the CIMS system used in this work is very similar to that described in the previous section as shown in Figure S2. For this experiment, the flow tube was humidified by a constant

amount of water vapor by passing a flow of 10 ccm of pure N$_2$ through a water bubbler kept in an ice bath in order to mitigate effect from varying humidity in ambient air.





### 2.3.1 Calibration sources

A permeation tube (KIN-TEK Laboratories, Inc.) was used as a calibration standard for formic acid. The permeation rate of the formic acid permeation tube was quantified by passing the output of the permeation tube in a flow of 50 sccm of $N_2$

through DI water. The resulting water solution of formic acid was analyzed using ion chromatography (Metrohm 761 Compact ICs). A permeation tube (KIN-TEK Laboratories, Inc.) was used as a calibration standard for $Cl_2$. To determine the emission rate of the $Cl_2$ permeation tube, the output was measured using a method described in detail by Kazantseva et al., 2002 and Chen et al., 2016.

The $ClNO_2$ standard was generated by passing a humidified flow of $Cl_2$ in $N_2$ through a bed of sodium nitrite ($NaNO_2$)

via the following reaction:

$$Cl_2(g) + NO_2(aq) \rightarrow Cl^-(aq) + ClNO_2(g), \hspace{3cm} (R4)$$

The resulting $ClNO_2$ was quantified by passing it through a heated quartz tube (150-230°C). $NO_2$ was generated via thermal dissociation of $ClNO_2$ and measured by a CAPS (Cavity Attenuated Phase Sift) $NO_2$ monitor (Thaler et al., 2011; Kebabian et al.,2008). The sensitivity for $ClNO_2$ was acquired by measuring the increase in the $NO_2$ concentration simultaneously with the

decrease in $ClNO_2$ signal in CIMS.

### 3 Results

### 3.1 Performance of LAS for the CIMS measurement of PAN

Performance of the LAS is compared to a standard 20 mCi radioactivity source by examining the detection sensitivity and the background signal level. Figure 3a and 3b show the sensitivity of both sources to $^{13}C$-PAN as a function of ion source

flow at total flow of roughly 2.7 slpm and 6.5 slpm respectively. In general, the sensitivity of the LAS is approximately 2 to 4 times lower than that of the standard source, whereas the activity ratio is roughly 13. Both sources give sensitivities of the order of tens of Hz pptv$^{-1}$ with limits of detection for PAN (LOD) of less than 2 pptv for a 2 minute integration (for a 23% duty cycle at m/z of 59 amu). Where LODs are estimated as 3 times the standard deviation of the background level. Maximum sensitivity was observed at flow tube pressures of 40 torr for both sources and total flows. We could undoubtedly gain more

sensitivity at flow tube pressures higher than 40 torr. However, we have found that higher pressures significantly enhance the risks of interference from unwanted secondary chemistry (Chen et al., 2016). The tests do show that the LAS is more sensitive to pressure than the standard source. In contrast, the sensitivity of the LAS for PAN is significantly less dependent on the ion source carrier gas flow compared to the standard source (Figure 3a). In particular, the LAS provides near maximum sensitivity at ion source flows a little greater than 1 slpm as illustrated in Fig. 3a. Thus, a low activity source may be preferable in some

applications, where reaching maximum sensitivity is not necessary (e.g., high ambient mixing ratios of targeting species), and the availability of carrier gas is limited.



### 3.2 Performance of LAS for the CIMS measurement of Formic Acid, $Cl_2$ and $ClNO_2$

Figure 4 presents the sensitivity comparison for the detection of formic acid, $Cl_2$ and $ClNO_2$. The trends presented were generally consistent with the PAN measurements. LODs were 1pptv, 1pptv and 0.5pptv, respectively, with an integration
time of 1 minute. The sensitivity of the LAS is approximately 3 to 5 times lower than that of the standard source. Maximum sensitivity for both sources was observed at a flow tube pressure of 40 torr for all three species. Similar to the measurements of PAN, the ratio of the LAS sensitivity to that of the standard source is greater at lower flows.

### 3.3 Application to Studies of the Ambient Atmosphere

The performance of the LAS source is illustrated in airborne observations of PAN during the Atmospheric
Tomography (ATom) field mission shown in Figure 5. Global-scale sampling of the atmosphere, profiling continuously over a span of altitudes, was conducted by an extensive gas and aerosol payload on board of the NASA DC-8 research aircraft. Figure 5a illustrates the performance of the low radioactivity source for the ATom instrument conditions along with the sensitivity values using a standard source. Figure 5b demonstrates flight data collected within a southern hemisphere biomass burning plume near New Zealand on Oct. 8th, 2017 (ATom-3 science flight #5), where gaseous reactive nitrogen ($NO_y$) is
dominated by PAN. Figure S4 shows the flight path coloured by the mixing ratio of PAN. Inside the plume intersects, the clear variation of PAN signal ($CH_3C(O)O^-$; 59 amu) is observed with a good signal to background noise ratio. Outside of the plume, PAN signal decreased simultaneously with the observed $NO_y$ mixing ratios. Clearly, the LAS can provide sensitive, fast time response detection of PAN from an aircraft platform.

### 3.4 Time dependence of sensitivity

As $^{210}Po$ ion sources age and lose activity the reagent signal decreases as well. In our experience, with a 20 mCi source you lose sensitivity at less than the decay rate of the $^{210}Po$. We typically observe a loss of sensitivity of about a factor of 2 in sensitivity after approximately one year. In order to estimate the time response of the sensitivity of the LAS we compared a fresh LAS to that of one that was decayed by approximately one $^{210}Po$ half-life ($\tau_{1/2} \sim 138$ days). The sensitivity of the LAS ( Figure S2) was found to decay about a factor of two after one half-life. This indicates that the sensitivity of the LAS
will decay faster than that of a standard source.

### 4 Discussion

The relative sensitivity of the LAS and the standard 20 mCi source is not in proportion to their activity ratios. The ratio of LAS sensitivity to activity is clearly higher than for a standard source. A likely explanation for much of this difference is the geometry of the sources. The LAS contains three cartridges mounted on a stainless-steel foil place in a QF40 nipple.
This gives an effective inner diameter of the ion source of ~3 cm. The standard source, NRD P-2031, has a more compact geometry with 1/4" NPT fittings on each end and an effective inner diameter of ~1 cm. The different inner diameters impact

ion source flow characteristics such as residence time and ion loss rates to the wall. For a given flow rate the larger ID LAS is expected to have a lower wall loss than the standard source. In addition, the larger ID of the LAS provides a longer path length for the emitted alpha particles. In short, the more open geometry of the LAS likely provides for more efficient transport of the
generated ions. This is also in accord with the observation that the LAS signal decays at approximately the same rate as the activity of the $^{210}$Po.

## 5 Conclusions

A low activity $^{210}$Po ion source with an initial activity of 1.5 mCi is a viable alternative to the larger activity sources typically used for I$^-$-CIMS. The sensitivity and detection limits of low activity ion source presented in this work are more than
adequate to measure PAN, Cl$_2$, ClNO$_2$ and formic acid in most environments. In applications that require maximum sensitivity, and the availability of carrier gas for the ion source flow is not limiting, a standard source is probably preferable. In field experiments that have limited or no radioactive storage the LAS may be a preferable option. The LAS may also be extended to other CIMS methods and reagent ions and will be a focus of our future ion source development.

## Data Availability

All of the data used in this manuscript is available upon request of the corresponding author. The field observations from the ATom campaign are available at: https://daac.ornl.gov/cgi-bin/dataset_lister.pl?p=39.

## Author Contributions

Young Ro Lee performed the majority of the experiments and wrote the manuscript with assistance from Greg Huey. Yi Ji helped perform the experiments to characterize the sensitivity of the compounds detected by adduct formation. David
Tanner assisted with all the experiments and performed the airborne measurements during the ATom field campaign.

## Competing Interests

The authors declare that they have no conflicts of interest.

## Acknowledgements

This work was supported by NASA grant NNX15AT90G. This publication was also partially supported by an EPA
STAR Grant R835882 awarded to Georgia Institute of Technology. It has not been formally reviewed by the EPA. The views expressed in this document are solely those of the authors and do not necessarily reflect those of the EPA. EPA does not endorse any products or commercial services mentioned in this publication and EPA Grant. The NO$_y$ data was provided by Thomas Ryerson, Chelsea Thompson, and Jeff Peischl.



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

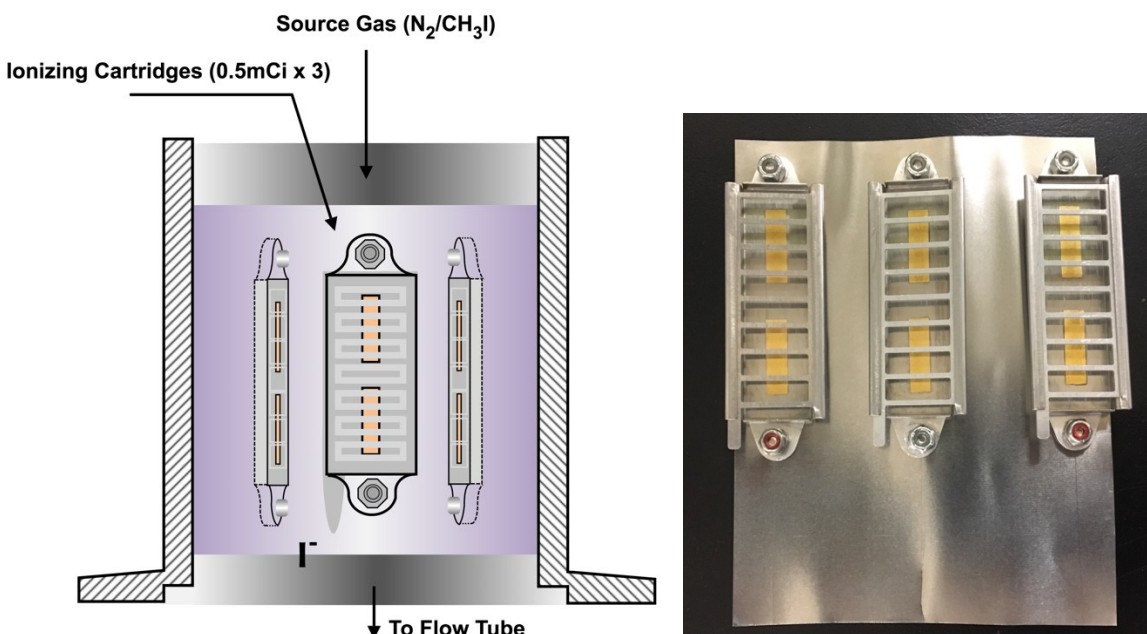

**Figure 1.** (a) Schematic of the LAS with three anti-static ionizing cartridges are placed in a standard QF-40 nipple. (b) Photograph of the $^{210}$Po cartridges mounted on a stainless steel foil.





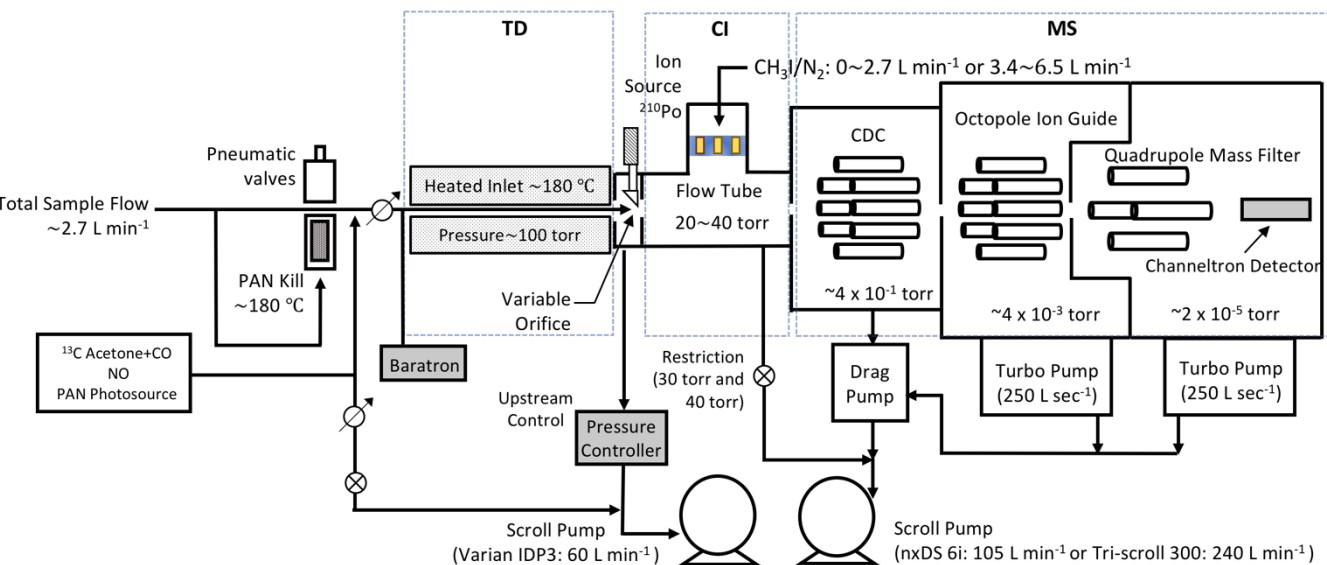

**Figure 2.** Schematic diagram of the TD-CIMS. The air is sampled through the TD (thermal dissociation) region and enters the flow tube through the AVO (automatic variable orifice).



**Figure 3.** The sensitivity (Hz pptv$^{-1}$) of the LAS and a standard source for PAN at different flow tube pressures and total flows, plotted as a function of ion source flow. (a) PAN sensitivity using the LAS, and (b) the standard source at total flow of ~2.7 slpm. (c) PAN sensitivity using the LAS, and (d) the standard source at total flow of ~6.5 slpm. Red circles, cyan squares and green triangles correspond to flow tube pressures of 20, 30, and 40 torr, respectively.

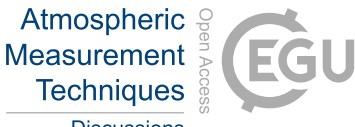





**Figure 4.** Comparison of the sensitivity (Hz pptv$^{-1}$) of the LAS and a standard source for formic acid (green) , Cl$_2$ (red) and
ClNO$_2$ (cyan) plotted as a function of ion source flow (slpm). Panels a-c show the sensitivity at flow tube pressures of 20, 30
and 40 torr at a total flow of 2.7 slpm (left) and 6.5 slpm (right).

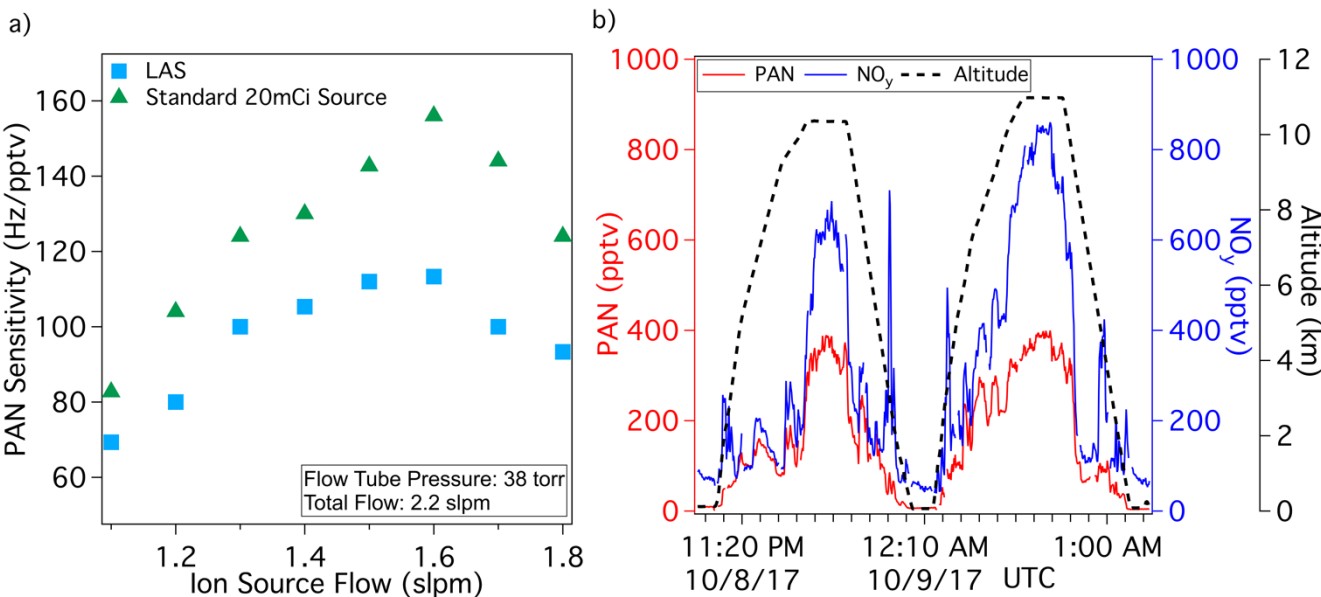

**Figure 5.** The performance of the LAS for PAN measurement during the ATom field mission. (a) The sensitivity comparison
of the LAS and a standard source for the ATom instrument conditions. (b) DC-8 flight data obtained during intersects of a
southern hemisphere plume. PAN (red trace) and NO$_y$ (blue trace) are plotted along with the pressure altitudes (black trace).
