# Peer review of "A low activity ion source for measurement of atmospheric gases by CIMS"

_Atmospheric Measurement Techniques, 2019_

## Referee Comment (RC1) · Anonymous Referee #1 · 2 Jan 2020

The authors present a new, low activity ion source for CIMS measurements. This is an important technical advance that could be advantageous for many researchers. The authors note that the sensitivity of the instrument does not scale with the activity, and that sufficient sensitivity can be achieved using the low activity source as demonstrated using field data. The paper is a well written technical manuscript.

A few short comments for the authors to consider:

Since the major advantage of the low activity source is the ability to transport it without HazMat training, it would be helpful for the authors to be more clear on what (if any) limitations exist for transporting the sources. Or more specifically, at what activity can a source be sent without HazMat training?

[Figure]

Section 3.1 provides a nice discussion of sensitivity, but almost no discussion on backgrounds which drive LOD in many CIMS instruments. It would be helpful to include a discussion of how the backgrounds changed between the standard source and the LAS and how that translates into LOD.

—————————————————

---

## Referee Comment (RC2) · Anonymous Referee #2 · 4 Jan 2020

Lee et al. tested a low activity radioactive ion source in a chemical ionization mass spectrometer. The manuscript is concise and written well. I only have minor comments (see below) which I am certain will be able to address before final publication.

General comments

(1) In several figures, the low activity source (1.5 mCi when new) was compared with the 'standard' source (20 mCi when new); however, these activities change over time due to the radioactive. In all figures, the age of the sources should be added.

(2) As far as I know, ion sources with activity of <10 mCi only require white (NON-RQ) shipping papers, which a 20 mCi reaches after ∼5 months (https://nrdstaticcontrol.com/images/returns/SHIPP2000.pdf). Would using a half-year

old 'standard' source be a viable alternative to the LAS described here?

Minor comments

line 9 - consider replacing "complications" with "regulatory burden"

line 11 - missing comma following (HCO2H)

line 18 - consider replacing "some field applications" with "short-term field deployments" or similar

line 60 - "49CFR 173.410; 49CRF 173.425, Table 4" What does this mean? Where is table 4?

line 75-113. Interference from peroxyacetic acid (and how it was minimized - see Phillips et al. Atmos. Chem. Phys., 13, 1129-1139, 10.5194/acp-13-1129-2013, 2013) should be mentioned.

line 82 -84. "PAN calibration standard" consider moving these 2 sentences to section 2.3.1 where the other calibration sources are described

line 100 - remove comma following et al.

line 112 - replace "detected" with "quantified"

line 119 - replace "as shown in Figure S2" with "and is shown in Figure S2"

line 121 - "effect from varying humidity". please state what is affected by RH (sensitivity?)

line 140 - "In general, the sensitivity of the LAS is approximately 2 to 4 times lower than that of the standard source, whereas the activity ratio is roughly 13" Are the values obtained with the LAS consistent with a standard source aged to an activity of 1.5 mCi? This is partially discussed later on but perhaps worth mentioning here.

line 149 - "may be preferable in some applications". Another example would be in polluted areas, where a LAS may be preferred also as the signals at m/z 59 may get

large (>10ˆ5)

line 137 - 151. "Performance of LAS". The results shown in Figures 3c and 3d are not talked about at all.

line 153. What ions were used to monitor formic acid, chlorine, and nitryl chloride? (should be stated here).

The sensitivities reported for chlorine and nitryl chloride are 2 orders of magnitude larger than reported by others (e.g., Mielke et al., 2011) who observed similar PAN sensitivity under conditions similar to panel (a).

line 165 "Figure S4" is called out before "Figure S3"

line 172 "less than the decay rate" Is this due to the generation of radioactive daughters?

line 174 "Figure S2" should be Figure S3.

Figure 3b - correct typo in axis caption (ion source fow)

Figure 3, caption. Please specify the ions monitored in the caption.

The difference between panels 3a and 3c isn't very clear. I suppose 3a was acquired at 2.7 slpm total flow and 3c at 6.5 slpm? Consider adding this information to textboxes to the figures (as was done in Figure 4).

Figure 4 - consider increasing the contrast between squares and circles as they're challenging to tell apart (e.g., replace solid with open circles).

Figure 4, caption. Please specify the ions monitored.

Figure S1c - 13C-PAN - is this m/z 61? Please clarify in the caption. "showing no obvious interference".

The signal is plotted on a logarithmic scale, so changes in intensity are difficult to spot to begin with. It appears to that there may some effect (factor of 2 perhaps) but it's

difficult to see. Please show these data on a linear scale.

To claim "no obvious interference" there would have to be some change in [NO] and [NO2]. Consider adding a scatter plot of the internal standard against [NO].

---

## Referee Comment (RC3) · Anonymous Referee #3 · 4 Feb 2020

This manuscript describes chemical ionization method using a lower radiation source activity of 1.5 mCi with three commercially available anti-static ionizing cartridges (2U500, NRD), as opposed to using a P-2021 at 10 mCi or P-2031 at 20 mCi. The ionizing cartridges are available to the general public and these devices are used here without any modification. Since 2U500 can be shipped and received without radiation source licenses, this improvement simplifies logistics in field campaigns. So this is a necessary step towards reducing the activity level of radiation in aircraft and field experiments. However, the authors should also mention that even this low activities (e.g., 0.5 mCi) are not exempted and in many states, they still require a radiation source license (even when used in the lab on the academic campus).

Please elaborate (Line 50) "For example, we have found that a significant fraction of

the 20 mCi ionizers have leaks on the body of the device."

---

## Author Comment (AC1) · 5 Mar 2020

Response to Referee 1

General Comments

1. Referee comment: "Since the major advantage of the low activity source is the ability to transport it without HazMat training, it would be helpful for the authors to be more clear on what (if any) limitations exist for transporting the sources. Or more specifically, at what activity can a source be sent without HazMat training?"

Author response: We have added more details on the transport regulations for the sources. Namely, the maximum activity per package to be shipped without HazMat training is 9 mCi (369 MBq), but this still requires shipping documentation. Ionizers

with activity below 5.41 mCi (200 MBq) can be shipped as an excepted package that only requires the UN number on the package label. The package still needs to meet the general design requirements of radioactive materials. We have added the maximum activity level for excepted package and referenced the code of federal regulations (CFR) in the revised manuscript.

2. Referee comment: "Section 3.1 provides a nice discussion of sensitivity, but almost no discussion on backgrounds which drive LOD in many CIMS instruments. It would be helpful to include a discussion of how the backgrounds changed between the standard source and the LAS and how that translates into LOD"

Author response: As the referee suggested, we have reported the range of background signals and LOD in Section 3.1.

---

## Author Comment (AC2) · 5 Mar 2020

General Comments

1. Referee comment: "In several figures, the low activity source (1.5 mCi when new) was compared with the 'standard' source (20 mCi when new); however, these activities change over time due to the radioactive. In all figures, the age of the sources should be added"

Author response: We have added the age of the sources to the caption of each figures.

2. Referee comment: "As far as I know, ion sources with activity of <10 mCi only require white (NON-RQ) shipping papers, which a 20 mCi reaches after ∼5 months. Would using a half-year old 'standard' source be a viable alternative to the LAS described

here"

Author response: The referee is correct in stating that an ionizer below 9 mCi can be shipped as "NON RQ" packaged of UN2911, which requires documentation and a UN label on the package. For excepted package, the activity level has to be below 5.41 mCi (200 MBq). However, it is very difficult for us to use an "old" source due to the regulations at our Institute.

Minor Comments

line 9) consider replacing "complications" with "regulatory burden"

We have modified the texts as suggested.

line 11) missing comma following (HCO2H)

We have added comma following (HCO2H).

line 18) consider replacing "some field applications" with "short-term field deployments" or similar.

We have modified the texts as suggested.

line 60) "49-CFR 173.410; 49CFR 173.425, Table 4" What does this mean? Where is table 4?

The code in the parentheses represents federal regulation about shipping and handling of radioactive materials. Table 4 describes the activity limits for limited quantities and instruments and is included in the code of federal regulation (49CFR 173.425).

line 75-113) Interference from peroxyacetic acid (and how it was minimized - see Phillips et al. Atmos. Chem. Phys., 13, 1129-1139, 10.5194/acp-13-1129-2013,2013) should be mentioned.

The isotopically labelled PAN calibration standard was added to air drawn through a QF 40 nipple filled with heated stainless-steel wool, which limits the interference from ambient PAA. In addition, the background measurement used for the ATom was performed by NO addition. We have added Phillips et al. 2013 to discuss possible interference from peroxyacetic acid.

line 82-84) "PAN calibration standard" consider moving these 2 sentences to section 2.3.1 where the other calibration sources are described.

We have modified the texts as suggested.

line 100) remove comma following et al.

We have modified the texts as suggested.

line 112) replace "detected" with "quantified"

We have modified the texts as suggested

line 119) replace "as shown in Figure S2" with "and is shown in Figure S2"

We have modified the texts as suggested.

line 121) "effect from varying humidity". please state what is affected by RH (sensitivity?)

We have modified the texts from "effect from varying humidity" to "variations in instrument sensitivity affected by ambient humidity".

line 140) "In general, the sensitivity of the LAS is approximately 2 to 4 times lower than that of the standard source, whereas the activity ratio is roughly 13" Are the values obtained with the LAS consistent with a standard source aged to an activity of 1.5 mCi? This is partially discussed later on but perhaps worth mentioning here.

We have added "Thus, it is likely that the initial activity of the LAS can result in higher sensitivity than the standard source aged to an equivalent activity level ($\sim$18 months), where the time dependence of sensitivity for both sources is discussed in section 3.4"

line 149) "may be preferable in some applications." Another example would be in pol-

luted areas, where a LAS may be preferred also as the signals at m/z 59 may get large (>105)

As the referee mentioned, the LAS may be a preferable option in polluted areas that have elevated levels of ambient PAN (e.g., fire plumes).

line 137-151) "Performance of LAS". The results shown in Figures 3c and 3d are not talked about at all.

We have added more discussion on the experiments using a larger scroll pump (Figure 3c and 3d).

line 153) What ions were used to monitor formic acid, chlorine, and nitryl chloride? (should be stated here). The sensitivities reported for chlorine and nitryl chloride are 2 orders of magnitude larger than reported by others (e.g., Mielke et al., 2011) who observed similar PAN sensitivity under conditions similar to panel (a).

We have added the ion signals monitored for formic acid (m/z 179), $Cl_2$ (m/z 197) and $ClNO_2$ (m/z 208). Absolute sensitivity presented in this work is specific to the experimental conditions and instrument parameters including transmission efficiency and declustering strength. For the configuration used for $I^-$ adduct CIMS, the collision dissociation chamber (CDC) was operated at a relatively lower voltage (-10 V), resulting in electric field < $\sim$22 V cm-1 than that for TD-CIMS experiments (-25 V).

line 165) "Figure S4" is called out before "Figure S3"

We have renamed the figures from "Figure S4" to "Figure S3."

line 172) "less than the decay rate" Is this due to the generation of radioactive daughters?

We think that at 20 mCi you are not limited by the activity of the source due to space charge effects.

line 174) "Figure S2" should be Figure S3

We have renamed the figures from "Figure S2" to "Figure S4"

Figure 3b and 3 caption) correct typo in axis caption (ion source fow). Please specify the ions monitored in the caption

We have corrected typo in axis caption and added the ions monitored in Figure 3.

Figure 3) The difference between panels 3a and 3c isn't very clear. I suppose 3a was acquired at 2.7 slpm total flow and 3c at 6.5 slpm? Consider adding this information to textboxes to the figures (as was done in Figure 4).

As the referee suggested, we have added textboxes to Figure 3 to eliminate any confusion regarding the different total flows.

Figure 4 and 4 caption) consider increasing the contrast between squares and circles as they're challenging to tell apart (e.g., replace solid with open circles).

We have increased the contrast between squares and circles by using lighter colors for circles. The ions monitored has been added in the caption.

Figure S1c) 13C-PAN - is this m/z 61? Please clarify in the caption. "showing no obvious interference". The signal is plotted on a logarithmic scale, so changes in intensity are difficult to spot to begin with. It appears that there may some effect (factor of 2 perhaps) but it's difficult to see. Please show these data on a linear scale. To claim "no obvious interference" there would have to be some change in [NO] and [NO2]. Consider adding a scatter plot of the internal standard against [NO].

We have added the ion monitored for 13C-PAN (m/z 61) in the caption. We have illustrated the data on a linear scale, and modified the texts from "showing no obvious interference" to "showing a minimal interference." Figure S4d has been added to exemplify relative sensitivity of PAN as a function of NO concentrations, showing less than 20% reduction in the detected ion signal for PAN by NO mixing ratios below 100 ppbv.

---

## Author Comment (AC3) · 5 Mar 2020

Response to Referee 3 (Referees' comments are italicized)

General Comments

1. Referee comment: "Since 2U500 can be shipped and received without radiation source licenses, this improvement simplifies logistics in field campaigns. So this is a necessary step towards reducing the activity level of radiation in aircraft and field experiments. However, the authors should also mention that even this low activities (e.g., 0.5 mCi) are not exempted and in many states, they still require a radiation source license (even when used in the lab on the academic campus)"

Author response: The 500 $\mu$Ci static eliminator cartridges are a generally licensed

device so don't necessarily require a site license. The degree of regulation varies by institution and state. For example, the LAS cartridge are exempted in the state of Georgia, but Georgia Tech requires them to be listed on their license as well. We have added "The 500 $\mu$Ci static eliminator cartridges are a generally licensed device. However, the degree of regulation is specific to the institutions and states in which they are deployed." to the revised manuscript.

line 50) Please elaborate "For example, we have found that a significant fraction of the 20 mCi ionizers have leaks on the body of the device."

Author response: In our experience, we have found that about half of the 20 mCi ionizers have vacuum leaks on the body of the device. They generally leak on the screws under the product label. In addition, we have found that leaks are possible near the ends of the ionizers where the central tube is mated to the pipe threads on the ends. We have added text noting the potential leaks.